# The Development of a Regional Tourism Destination Competitiveness Measurement Instrument

Tanya Rheeders  and Daniel F Meyer *

College of Business and Economics, University of Johannesburg, Johannesburg 2006, South Africa
* Correspondence: dfmeyer@uj.ac.za

**Abstract:** The importance of the tourism sector has been highlighted and featured in various studies indicating not only the economic but also social and environmental benefits. There is a need for a measurement instrument for regional tourism destination competitiveness. This measurement instrument could gauge a destination's regional potential for tourism development and competitiveness; and be able to compare regions. To conduct an instrument development and validation, both PLS-SEM for confirmatory factor analysis and SPSS were utilised for exploratory factor analysis. A purposive sampling approach were used for both study areas, Sedibeng and Fezile Dabi district municipal regions, in which pilot studies were executed through a survey between July to September 2020. The reliability of the measurement instrument was confirmed with Cronbach's Alpha ($\alpha$) for both samples having a value above 0.70. The EFA confirmed the validity of the measurement instrument for the three-dimension and 16-items of the measurement instrument. This study recommends using the measurement instrument as a practical tool to analyse regions regarding the development and competitiveness of a tourism destination compared to other destinations.

**Keywords:** factor analysis; measurement instrument; South Africa; tourism destination competitiveness

## 1. Introduction

The level of regional competitiveness of a tourism destination contributes to the economic development process (Ozer, Küçüksakarya, & Maiti, 2022) [1]. In reviewing the literature, it was found that there is still a need for research in regional tourism destination competitiveness as there is no extensive evidence of the existence of a measurement tool for regional tourism destination competitiveness in an empirical format, especially to compare regions. The main aim of this research paper was to fill this gap in the tourism research. Research by Lopes, Muñoz and Alarcón-Urbistondo (2018) [2] identify problems with the measurement of regional tourism destination competitiveness by stating that most organisations' measurements have been implemented on a national level with very few focusing on a regional level. The development of an empirical measurement instrument is needed on a regional and local level, which will assist in comparing the region's tourism development and competitiveness. The purpose of the research is therefore to develop an empirical measurement instrument that assists in the determination of a region's tourism destination competitiveness.

As indicated by the Travel and Tourism Competitiveness Index by the World Economic Forum (WEF), a lack of tourism destination competitiveness is one of the hindrances South Africa faces in addition to suitable education, employment, health services and other basic human needs. Tourism and especially international tourism have been identified as a solution to these challenges (Du Plessis, Saayman & Van der Merwe, 2015) [3]. The OECD (Organisation of Economic Co-operation and Development) (2019) [4] links competitiveness with these challenges by stating that a region's tourism destination competitiveness should increase if the labour force in the tourism sector is properly trained, leading to an increase

in their productivity. Improving the labour force in the tourism industry could improve competitiveness within a specific region.

The development of the tourism sector should be considered a priority for most countries due to the advantages arising for the local community and the economy. These advantages could be beneficial not only to the tourism industry but also to tourism-related businesses such as adventure activities, accommodation facilities, conference and wedding venues, food and beverage facilities, souvenir shops, tour agencies and guides, and transportation services. Shahzad, Shahbaz, Ferrer and Kunmar (2017) [5], postulate that community members can benefit via employment opportunities. The Department of Tourism (2020) [6] argues for the improvement in employment opportunities, as novel business owners can easily enter the tourism industry. This links to the benefits stated by Meyer and Meyer (2015) [7] that job creation has on the economy and local people.

Even though the tourism destination-related models by authors such as Crouch and Ritchie (1999) [8] and Dywer and Kim (2003) [9] are the most agreed-upon models, some important determinants crucial for the progress of a tourism destination, such as technology and political environment, were excluded. Furthermore, these models are conceptual models, which allows for a gap in the literature, focusing on a practical measurement instrument. The models such as the WEF's TTCI focus on the degree of tourism destination competitiveness on a national level and the results for TTCI are only published every second year. The tourism industry is fast-paced and dynamic; thus, regular analysis is required to investigate regional tourism destination competitiveness. This brings forth the gap in the on a regional level for a measurement instrument (Abdel-Basset, Mohamed & Smarandache, 2018) [10].

The necessity for an empirically measurable instrument on a regional level was evident throughout the review of the literature. During the research, it was found that the existing empirical model, TTCI (Travel and Tourism Competitiveness Index), is developed on the same weighting scale, which states that the determinants have equal importance in determining tourism destination competitiveness. According to Martín, Mendoza and Román (2017) [11], the importance of weighting values of determinants should truly represent its importance through theoretical and numerical properties. This new measurement instrument contributes to research by investigating the different priority and importance weights of the dimensions and determinants included in the model.

## 2. Literature Review of Measurement Instrument Development explaining Tourism Destination Competitiveness

Within the global economy, competitiveness is increasingly becoming a requirement to remain successful (Lustický & Stumpf, 2021) [12] (Shariffuddin Azinuddin, Hanafiah & Zain, 2021) [13]. This is not different for tourism destinations (regions). In order for a tourism destination to remain competitiveness is should (i) ultimately attract tourist and/or visitors (ii) grow with international globalisation (iii) provide a unique experience (Shariffuddin Azinuddin, Hanafiah & Zain, 2022) [13]. The tourism competitiveness of a region could lead to a range of benefits. Rodríguez, Florido and Jacob (2020) [14] agree that the tourism sector is an important contributor to economic growth and job creation. Infrastructure development and tax generation are amongst the economic benefits of an increase in tourism development (Cavalheiro, Joia & Cavalheiro, 2020) [15]. In addition, economic development is known to be a benefit for tourism development (Rodríguez, Florido & Jacob, 2020) [14]. The increase in technology, skills development and higher human capital contribute to economic development in a region.

According to Madanaguli, Srivastava, Ferraris and Dhir (2022), [16] in recent years countries focused not only on the financial gains of economic activities but also focusing on the environmental aspect thereof and as such sustainable tourism development is a key objective of tourism destinations. They main negative consequence of extreme tourism develop is resource depletion (Rodríguez, Florido & Jacob, 2020) [14]. Some regions do not have the capacity to compensate for the increase rise in economic and social activities. Social

Corporate Responsibility comes into play as companies and regions takes into account how business practices, policies, growth and enhancements influence the region's environment (Madanaguli, Srivastava, Ferraris & Dhir (2022) [16]. The two key components that are influences by tourism development is the environment and the community members. Therefore changes, enhancements and improvements of the region should take into account the influence it will have on the environment and community members.

The tourism-led growth hypothesis states the importance of tourism development in the growth of a region's economy (Xia, Doğan, Shahzad, Adedoyin, Popool & Bashir, 2022) [17]. According to Pérez-Montiel, Asenjo and Erbina (2021) [18], the tourism-led growth hypothesis explains that the increase in tourism development leads to an increase in economic growth. This theory has been proven by various studies (Balaguer & Cantavella-Jordá (2002) [19]; Pérez-Montiel, Asenjo & Erbina, 2021 [18]) which adds to its validity as a premise for this study. The use of the tourism destination measurement instrument will be mainly as an indicator of which areas in a region can be used with opportunities of development, which are the strengths a region can build on, which are the weaknesses a region needs to minimise and any threats to development that should be anticipated in the future. The end goal is to increase tourism development of a region keeping in mind the needs of the environment and community members.

The determinants of tourism destination competitiveness include the following: According to Lo, Mohamad, Chin and Ramayah (2017) [20] natural and cultural resources can boost a tourism destination's degree of competitiveness through sensible resource utilization and efficient management. A tourist destination is more likely to be successful with promoting of tourism expenditure when it is known for stunning scenery and intriguing features, claim Andrades and Dimanche (2017) [21]. The quality and availability of tourist attractions were determined to be crucial factors in three of the four locations studied by Csapó, Habil, Pintér and Aubert (2016) [22] to build a tourism destination. According to Jaafar, Rasoolimannesh and Lonik (2015) [23], the tourism sector can be an excellent place for a small firm to get started as it requires less start-up funds. The development of job prospects is a benefit of tourism growth (Crouch & Ritchie, 1999) [8]. However, in order for tourism locations to develop successfully, quality labor is required. Opportunities for economic diversification are facilitated by the effective management of infrastructure that supports tourism activities (Jovanović & Ivana, 2016) [24]. The effectiveness of structures and infrastructure improved the perceptions of a tourist location, according to a study by Csapó, Habil, Pintér and Aubert (2016) [22]. First, it is crucial for regional governments to ensure the growth of the tourism sector, according to Kubickova and Hengyun (2017) [25]. Second, regional governments and authorities are unable to successfully intervene in the tourism industry and risk deterring tourism growth through unjustified controls.

The concept of "measurement instrument" is used synonymously with concepts such as index, scale or tool in this study. A measuring instrument or scale is a measure that pools the values of various items (indicator variables) into a combined measurement (Straus & Wauchope, 1992) [26]. However, the validation of a measurement instrument can be complicated. Van Peer, Hakemulder and Zyngier (2012) [27] state that the types of the measurement instrument are impacted by (i) the quality of data received, (ii) the determinations that are done, and (iii) the statistical examination performed. Scales have been included as a component of the measurement instrument. Scales were used in the development and testing phases of the measurement instrument. In the development phase, a scale is used by subject experts in pre-testing to determine the average importance of the determinants in terms of tourism development. When testing the measurement instrument by pilot studies in the regions, respondents were required to identify the level of importance of a tourism development determinant in a specific region.

There exist various studies that aim to development a measurement instrument for tourism destination development and/or competitiveness (TDC) in some form. In 2010, Ritchie and Crouch (2010) [28] studied the development of a model that measure TDC through conducting qualitative interviews. This study identified five main groups (i)

qualifying and amplifying determinants, (ii) destination policy, planning and development, (iii) destination management, (iv) core resources and attractors and (v) supporting factors and resources) with various sub-factors that have an influence on TDC.

In the study of Hanafiah, Hemdi and Ahmad (2016) [29], the goal was to develop a performance-based model of TDC based on competitiveness theory. This study builds on studies such as Buhalis and Spada, (2000), Health (2003) and Mazanec et al. (2007) to name a few developing a conceptual model. Hanafiah et al. (2016) [29] state that there is still a need for research in the determinants of TDC due the complicatedness of the tourism sector. This study undertaken a conceptual approach in identifying factors that contribute to TDC.

Selim, Abdel-Fattah and Hegazi (2021) [30] investigated the smartness and competitiveness of the factors namely, attractiveness of heritage destinations by developing a composite model. By analysing "key performance indicators" a mix-method approach utilising EFA and CFA tests to determine the reliability and validity of the proposed composite index. A purposive sampling approach identified historians, history consultants, project managers and architects of historical sites to complete the survey. This study also has dimensions and factors within to describe heritage attractiveness in a tourism destination.

The study of Sul, Chi and Han (2022) [31] developed a measurement model of TDC through an empirical method. These authors are in agreement with the previous study *of* (van der Schyff, 2021) [32] and Rheeders (2022) [33] and the current study, that the different determinates are complex and diverse and that a generic model of measurement instrument could not be applied to adequately measure TDC. Sul et al. (2022) [31] methods also used a survey approach utilizing tourism-related managers as expected to complete a survey via convenience sampling. Conversely, this study (Sul et al. (2022) [31]) solemnly focuses on the business environment linking this to the competitive advantage theory, whereas the current study takes various social, political, environment and business environments into account. Sul et al. (2022) [31] also used Exploratory Factor Analysis (EFA) to measure internal reliability and Confirmatory Factor Analysis (CFA) for validity of the model.

Various studies attempt to describe TDC through measurement instrument development. There studies are important but takes a different approach to the current study. It is therefore important that this study is undertaken, as it sets out to develop a comprehensive measurement instrument that include determinants in social, economic, political and environmental aspects in providing an empirical instrument.

### 2.1. Scale Development Process

Due to the complexity of scale development, numerous methods or techniques can be utilised in developing a tourism destination measurement instrument (scale). The validity of the construct of a scale can be investigated using various methods, including these techniques, as listed in Table 1.

**Table 1.** Validity testing of a scale (measurement instrument).

| Type of Analysis | Existing Methods and Techniques | Methods used in the Study |
|---|---|---|
| Reliability | Cronbach's Alpha, Composite reliability (CFA & EFA) | x |
| Convergent validity (construct) Discriminant validity (construct) | Factor analysis Principal Axis Factor | X |
| Uni-dimensionality (construct) Nomological validity (construct) | Factor loadings and comparison between variances Correlation between scales | X |

**Table 1.** *Cont.*

| Type of Analysis | Existing Methods and Techniques | Methods used in the Study |
|---|---|---|
| Invariance Model fit | Fit indices (CFA) Modification of indices, standarised residuals, Squared multiple correlations fit indices. | |
| Factor analysis | Barlett's test of Spherity and Kaiser-Meyer-Okin's measure of sampling adequacy | x |
| Factor structure | Eigenvalues | X |

Source: Van der Schyff (2021) [32].

**Reliability** can be predicted through Cronbach's Alpha ($\alpha$). Sharma (2016) [34] postulates that this is used if there is more than one item, indicating if there exists coherence between the values indicated by respondents. Cronbach Alpha uses the estimate or determines the internal consistency that is associated with the scores from a scale. If there is no consistency, the scores on the scale will not be reliable. Patel (2015) [35] states that it would be acceptable if Cronbach's Alpha ($\alpha$) exceeds 0.70. Therefore, indicating if there exists coherence between the values indicated by respondents and that the reliability criteria have been met. According to Vaske, Beaman and Sponarski (2017) [36], Cronbach's Alpha can be influenced by the number of items on the scale (measurement instrument), dimensionality and the inter-correlation of the items in the construct (scale, measurement instrument). **Discriminant validity** is the result when no redundant items in the construct or scale exists (Ahmad, Zulkurnain & Khairushalimi, 2016) [37], leading to the "uniqueness of the construct" (Hashim, Mukhtar & Safie, 2019) [38]. The validity of a scale is necessary and achieved by discriminant validity (Franke & Sarstedt, 2019) [39] through (1) factor analysis (2) Principal axis factor (PAF) with a direct quartimin oblique orthogonal rotation. Factor loadings are generally accepted to explore the discriminant validity of the construct (Hashim, Mukhtar & Safie, 2019) [38]. **Uni-dimensionality** is the existence of one construct explained by a variety of items (Hattie, 1985) [40]. The development of measures usually includes more than two items explaining the construct, this investigates the relationship between these items. Anderson and Gerbing (1988) [41] postulate that an analysis of the composite score provided information as to whether or not the measure can be accepted. During the development of a measurement instrument, the items of which the construct comprised was tested by CFA. The CFA is used to test the unidimensionality as a method to "refine the scale", testing the construct. Factor loadings, cross-loadings and comparison between average variance extracted and squared correlation between each pair of constructs (shared variance test). The number of determinants that load onto each other should be known before further analysis. **Factor analysis** through (i) Bartlett's test of Sphericity is a hypothesis that states that the correlation matrix is known as the identity matrix. Therefore, the items on the scale are not related to one another and are not suitable for use, and (ii) Kaiser–Meyer–Olkin's measure of sampling adequacy provides the proportion of variance for the items. If the Kaiser–Meyer–Olkin's results are higher than one, the factor analysis is valuable. **Factor structure:** The Eigenvalues are used in dimensionality analysis (Chilcot, Guirguis, Friedli, Almond, Davenport, Day, Wellsted & Farrington, 2017) [42]. This study utilised factor analysis, factor structure and reliability techniques to validate the measurement instrument for regional tourism destination competitiveness successfully.

*2.2. Best Practice Principles in Scale (Measurement Instrument) Development*

The best practice principles, recommendations, guidelines and procedures for scale development from four important researchers are described in the following section. Throughout the research process, it has been found that the studies of Churchill (1979) [43], Hinkin's (1995) [44], Rossiter (2002) [45], DeVellis' (2003) [46] and Worthington and Whittaker's

(2006) [47] are seen as the main theories/ frameworks for the development of scales. Starting with the oldest to newest, these studies are explained in the following:

**Churchill's (1979) [43]** framework for the development of measures for constructs: According to Kock, Josiassen and Assaf (2019) [48], the development and application of measurement tools to quantify constructs, commonly referred to as "scales", are essential to knowledge creation within the social sciences. Six procedures are used to execute the procedure for the development of a measure successfully, as advocated by Churchill (1979) [43]:

- Procedure 1: Specify the domain of the construct. Through investigating the literature field, what is to be measured,
- Procedure 2: Generate a sample of items and data collection. The items can be identified through literature reviews, previous research including theories and questionnaires. Not all the items that have an impact on the construct must be used, but only a sample of the most significant items. This should give knowledge regarding which items influence the construct,
- Procedure 3: Purify measure and data collection. This is executed utilising factor analysis and Cronbach's Alpha coefficient. The factor analysis indicates the features describing the construct. The Alpha coefficient is used to investigate the internal consistency. This theory states that each item has a different significance in determining the construct,
- Procedure 4: Assess reliability. The face and content validity tests are used to test reliability. The Alpha coefficient can test the reliability of the measure. The higher Alpha value indicates that the items are stable and relevant in describing the construct. This is, therefore, a crucial statistical analysis,
- Procedure 5: Assess validity. The validity analysis ultimately indicates whether or not the construct is successfully and adequately presented. Moreover, discriminant validity is valuable. EFA was used to identify the dimensions using IBM SPSS. CFA was used to test for reliability and validity using SmartPLS,
- Procedure 6: Develop norms. The "raw score" resulted from the use of the measure. This raw score should be translated as the discussion of the level of measurement.

A study by **Hinkin** in 1995 investigated the development methods of a total of 277 scales between the time-period 1989 and 1994. Hinkin (1995) [44] made a model of the three stages of relevant steps within each phase for scale development:

- Stage 1: Item generation,
- Stage 2: Scale development,
- Stage 3: Scale evaluation.

**Rossitier (2002)** [27] suggests a substitute "procedure" that could be used when developing a measurement instrument. It utilised the C–OAR–SE (construct definition, object classification, attribute classification, rater identification, scale formation, enumeration, and reporting). This method considers reasonable opinions and consensus between experts in the field. This measurement only requires content validity as Rossiter (2002) [45] believes that the construct and predictive validity test is unsuitable for measuring a measure. Rossiter (2002) [45] also critiques Churchill's (1979) [43] framework as it only forms a part of the C–OAR–SE method. In addition, Rossiter (2002) [45] is opposed to the view that the scale development framework and the importance of factor analysis reliability testing could result in the appropriate scale describing a construct through uni-dimensionality. The identified steps in the development of a scale in the Rossitier (2002) [43] model of scale development are;

- Step 1: Construct definition: Give the construct definition and outline the scale's objectives,
- Step 2: Object classification,
- Step 3: Open-ended interview questions attribute classification to the sample frame. It is also necessary to categorise the object. Produce the items that denote the object,

- Step 4: Construct definition should be set out,
- Step 5: Rater (respondents) identification: Raters and the individuals conducting tests. This could include experts in the field,
- Step 6: Scale formation is used to unite the items and objects for the scale. To determine the adequate rating scale for the items that can measure open-ended questions. The rater's sample requires pre-testing,
- Step 7: Enumeration regards the implication of the scale. This is achieved by utilising index and average values to achieve a total score. For example, this could be a scale on a range from 0 to 10.

**Devellis (2003)** [46] formed a ten-step scale development process which was also the selected recommendation of Worthington and Whittaker (2006) [47]:

- **Step 1:** Determine what should be measured: the purpose of the measurement instrument should be clear. The investigation into theory could create a framework or reference to the objective of the measurement instrument,
- **Step 2:** Pooling the items characterising the construct: Items should be selected based on relevance to the construct. Starting with a larger number of items identified from step 1, items undergo a reduction process. The most important items with high relevance to the construct are selected, whereas items with little relevance to the construct will be removed,
- **Step 3**: Decide on the layout of the measurement. Concise and short to the point questionnaires are preferable,
- **Step 4:** Review item pool by experts. Make use of subject experts to give input regarding the relevance and quality of the items selected as a measurement of the construct. This is also a means to perform a content validity analysis. The face validity should be analysed by this process, investigating the clarity, to reduce redundant items. The significance of the items needs to be carefully analysed by the experts as it directly relates to the relevance of the items,
- **Step 5**: Validation of items by convergent and discriminant validation methods. The items that relate to the construct and those that give the complications are identified,
- **Step 6:** Administer items to sample. The adequate sample size is between 150 and 200, and a total of 300 are usually accepted. After identifying the relevant and validated items that adequately describe the construct, the final creation of the construct should be executed,
- **Step 7:** Evaluate items. The EFA technique can be made use of. The sampling method can include purpose sampling and a combination of purposive and convenience sampling. The CFA, goodness-of-fit index and model fit could be used for analysis. Factor analysis is used to determine the pooling or itemised groups constitute a unidimensional factor. The coefficient Alpha of reliability is also used to determine the quality of a scale,
- **Step 8:** Improve the scale length—reduction of the scale by use of specific criteria. The length is the scale, and the covariation impacts the Alpha mentioned above. It should be noted that a short scale simplifies the process for respondents to complete the questionnaire, whereas longer scales are more reliable. A balance in the length of the scale should be reached.
- **Step 9:** Cross-validation scales can be useful in instances where changes to the scale were made during the development process,
- **Step 10:** Develop norms for the scale: Norm development should be clearly set out to assist with the score explanation.

## 3. Materials and Methods

Tourism destinations are complex environments (Martín, Mendoza & Román, 2017) [11] and have various interlinking networks and industries. The progress in technology, the rapid adjustments of tourist requests and internationalisation add to the complexity of the ever-changing worldwide tourism industry. As a result, a multidimensional measuring

instrument should be developed to best analyse the level of tourism destination competitiveness at a regional and local level. To improve tourism-related facilities within a region, focus should be on not only the progress of structures but a variety of facilities and resources. This stresses the requirement for a measurement instrument assisting with analysing the performance of the tourism industry at regional and local levels, whereas previous research focuses more on an international level and comparing countries (Baggio, 2018 [49]; Boroomand, Kazemi & Ranjbarian, 2019 [50]. The study methodology used a functionalist approach. In order to develop a measurement instrument, various stages are required to complete.

Stage A: Literature review: Creswell (2014) [51] states that after understanding the research problem, it is crucial to do an extensive review of the literature. An extensive literature review was conducted on the determinants of tourism destination competitiveness in the study of Van der Schyff (2021) [32], and a list of determinants was selected as contributors to the success and development of a tourism destination on a regional level.

Stage B: Development of an instrument: Taking into account the multiple models of scale development, the steps (given as phases) of development and testing of the measurement instrument of tourism destination competitiveness was dealt with within the following section. In this study, a combination is used by Churchill's (1979) [43] and Hinkin's (1996) [44] recommendation of scale development to develop and validate the measurement instrument of tourism destination competitiveness. Although the development and testing of the measurement instrument follow the recommendation made by Churchill (1979) [43] and Hinkin (1995) [44] the phases used do not correlate point to point; however, some of the recommendations and steps were included within the phrases given. This methodological approach is custom to the study's objectives although it follows the recommendations of previous main frameworks.

- **Phase 1:** Identification of the construct domain– an investigation into determinants of TDC: The construct domain developed "tourism destination competitiveness measurement instrument" on a regional level tourism in the Sedibeng and Fezile Dabi district municipalities that form part of Gauteng province and the Free State province, respectively.
- **Phase 2**: **Determinants** selection: Item generation was performed through existing literature and the categorisation of items into determinants and dimensions. A literature review and previous research (Van der Schyff, 2019) [32] were used as a starting point for determinant selections on which the measurement instrument's development was based. Furthermore, existing models of tourism destination competitiveness were analysed to develop a comprehensive measurement instrument.
- **Phase 3:** Pre-testing: The initial data collection and purification by using expert validation, pilot testing and scale refinement, modification and finalisation were done.
- **Phase 4:** Adjustment and finalisation of the measurement instrument: Subsequently, there was the pre-testing phase. All inputs and recommendations from industry and subject experts were carefully taken into account and considered to ensure the best possible development of the measurement instrument.
- **Phase 5**: Measurement instruments' index calculation: The index value of each dimension and determinant were developed by use of the importance weights through the following formula:

$$\text{Index value} = \frac{Determinant\ of\ group\ value}{3.81\ (largest\ weigth\ value)}$$

The index value, therefore, indicated the importance of each determinant and dimension in achieving tourism destination competitiveness. The index value would be multiplied by the performance rating to produce a final tourism performance value.

- **Phase 6:** Questionnaire design: The rationale behind the use of a questionnaire was to collect the opinions of respondents active in the tourism industry. According to

Brandon (2011) [52], the questionnaire is acceptably used to collect information on respondents regarding specific areas.

- **Phase 7:** Pilot study: After the questionnaire was designed, the pilot study was performed to evaluate the performance of tourism destinations in terms of their competitiveness in being thriving tourism destinations. The pilot study used closed-ended questions, as respondents were asked to select a ranking position for each determinant and group on a scale. Trafford and Leshem (2008) [53], state that even though open-ended questions lead to a more detailed answer, closed-ended questions could be used to have brief and to-the-point answers. The open-ended question needs more thought, whereas close-ended questions are easier to answer, even though the questionnaire questions are closed-ended. In all, 320 questionnaires were completed for the district municipalities of Sedibeng and Fezile Dabi. This follows the 10:1 ratio-for each variable, ten questionnaires were completed for each district municipality. The questionnaires were either completed manually on a paper form or electronically on a link and document.

To collect the data, face-to-face, telephonic and mail correspondence were used. A purposive sampling approach is followed in collecting the questionnaires in the pre-testing and a pilot study. In the case of the pre-testing, industry and subject experts were required to give input as they knew this field. In the case of the pilot testing in each district municipality, (i) community members/tourists, (ii) tourism-related businesses, or (iii) government organisations within the district municipality have been selected as they know the performance of the determinants in the district municipality. The measurement instrument's validation and reliability were tested using EFA (exploratory factor analysis) to identify the dimension using IBM SPSS and CFA (confirmatory factor analysis) using SmartPLS. The following section discusses the test performed. Table 2 gives a summary of the statistical analysis of the measurement instrument of tourism destination competitiveness (development and testing).

**Table 2.** Summary of statistical analysis for the development of the measurement instrument.

| Type | Analyses | |
|---|---|---|
| Program | Smart-PLS3 | SPSS 28 |
| Model | PLS-SEM | |
| Analysis | CFA | EFA |
| Objective | Structural validity | Discriminants validity and reliability |
| Test | Factor loadings (composite/convergent reliability) AVE Cronbach's Alpha | Barlett's Test of Sphericity Kaiser-Meyer-oklin, Cronbach's Alpha |

Source: van der Schyff (2021) [32].

*Statistical Analysis for Instrument Development*

The factor analysis is a data reduction procedure analyzing the relationship between variables and identifies fewer variables than explaining these correlations or relationships in the form of (i) principal component analysis and (ii) common factor analysis. These analyses are important as various factors (determinants) are identified throughout previous studies (literature) that makes the use of a measurement instrument difficult. In addition, the analysis also presents the link between tourism development and/or competition to the determinants. For PLS–SEM, this study used SmartPLS software to investigate complex interconnections between variables (Sarstedt & Cheah, 2019) [54]. The rationale for using PLS–SEM was the ability to investigate complex models (Olya, 2017) [55]. Therefore, SmartPLS was used in this study to investigate the determinants of tourism destination competitiveness. With the PLS–SEM, the discriminants validity is given through the

AVE (average variance extracted). The AVE gives a construct's average variance and the measures (items or determinants). The current study made use of EFA and CFA, which utilises the following:

**Factor loadings**: To test the discriminant validity, the AVE values should be higher itself than any other construct (Hashim, Mukhtar & Safie, 2019) [38]. Identification of the factors' loadings, load onto the factors (determinants), and the SEM (structural equation modelling) can be performed for structural validity. **Average variance explained**: Average variance explained is used to test for convergent validity (Janadari, Sri Ramalu & Wei, 2016) [56]. The item loadings also indicated convergent validity. Janadari et al. (2016) [56] indicate that the AVE value should exceed 0.5, indicating that the value of the construct explained the variance of the items (determinants). **Cronbach's Alpha**: The internal reliability associated with the scores from a scale were predicted using Cronbach's Alpha ($\alpha$) (Hashim et al., 2019:4) [38]. According to Bryman and Cramer (2009) [57], internal reliability is a measurement used if there is more than one item and therefore applicable to the study. Patel (2015) [35] states that it would be acceptable if Cronbach's Alpha ($\alpha$) exceeds 0.70.

## 4. Results and Discussion

This section consists of the empirical findings from the development of the measurement instrument through validity and reliability testing for the measurement instrument. A tourism destination competitiveness measurement instrument was developed by means of the following phases based on the methods and processes as listed in the literature review section:

**Phase 1**: Identification of the construct domain– an investigation into determinants of TDC: The construct domain developed a „tourism destination competitiveness measurement instrument" which was tested on a regional level in the Sedibeng and Fezile Dabi District Municipalities that form part of Gauteng province and the Free State province, respectively.

**Phase 2**: Determinants selection: Item generation was performed through existing literature and the categorisation of items into determinants and dimensions. A literature review and previous research (Van der Schyff, 2021) [32] were used as a starting point for determinant selections on which the measurement instrument's development was based. Moreover, existing models of tourism destination competitiveness were analysed to develop a comprehensive measurement instrument.

**Phase 3**: Pre-testing: During Phase 3, the initial data collection and purification using expert validation, pilot testing and scale refinement, modification and finalisation was performed. Priority results of dimensions and determinants of tourism destination competitiveness are presented in Table 3.

**Table 3.** The priority values of selected determinants.

| Dimension or Determinant | Average Priority Value | Priority Rank |
|---|---|---|
| **1. Resources** | 1.74 | 2 |
| 1.1. Natural resources and strategic location | 1.81 | 1 |
| 1.2. Historical and cultural resources | 3.42 | 3 |
| 1.3. Technology, innovation and communication | 3.81 | 4 |
| 1.4. Entrepreneurship, the business community and workforce | 4.33 | 2 |
| **2. Infrastructure** | 1.71 | 1 |
| 2.1. Health and education facilities | 5.17 | 5 |
| 2.2. Accommodation facilities | 3.16 | 1 |
| 2.3. Transportation facilities | 3.58 | 2 |
| 2.4. Sport and recreation facilities | 5.74 | 6 |
| 2.5. Food and drink facilities | 4.32 | 4 |
| 2.6. Essential services | 3.97 | 3 |

**Table 3.** *Cont.*

| Dimension or Determinant | Average Priority Value | Priority Rank |
|---|---|---|
| **3. Enabling environment and authorities** | 2.55 | 3 |
| 3.1. Public–private partnerships | 5.35 | 6 |
| 3.2. Safety and security | 2 | 1 |
| 3.3. Government spending and efforts | 3.99 | 4 |
| 3.4. Local leadership and political stability | 3.77 | 2 |
| 3.5. Red tape limitation | 3.70 | 2 |
| 3.6. Macro–economic environment | 4.58 | 5 |

Source: Van der Schyff (2021) [32].

**Phase 4**: Adjustment and refinement of the measurement instrument: Industry and subject experts were consulted in the pre-testing providing the following recommendations: In addition to word documents, have online accessible questionnaires; reduce the number of determinates to guarantee easy questionnaire completion by respondents. Therefore, the two determinants "government spending on tourism and marketing efforts" and "sustainable tourism policies and destination management" were combined as "government spending and efforts"; combined education facilities to "health and education facilities"; communication facilities were moved to the dimension "resources" with "technology and innovation"; relocating "strategic location" as a factor for the determinant "natural resources as it is more appropriate. As a result, the measurement instrument had 16 determinants within the three dimensions explaining tourism destination competitiveness.

**Phase 5**: Calculation of index value: The pre-testing phase of the study was performed by industry and subject experts who assisted in the calculation of the index by providing the importance weighting each dimension and determinant. Table 4 provides the average importance weighting and the calculated index values for the tourism destination competitiveness measurement instrument by industry and subject experts.

**Table 4.** Importance of weight results for dimensions and determinants of tourism destination competitiveness.

| Dimension or Determinant | Average Weight Value | Index Value |
|---|---|---|
| **1. Resources** | **3.55** | **0.9317** |
| 1.1. Natural resources and strategic location | 3.34 | 0.8766 |
| 1.2. Historical and cultural resources | 3.16 | 0.8294 |
| 1.3. Technology, innovation and communication | 2.95 | 0.7742 |
| 1.4. Entrepreneurship, the business community and workforce | 2.87 | 0.7533 |
| **2. Infrastructure** | **3.45** | **0.9055** |
| 2.1. Health and education facilities | 2.74 | 0.7192 |
| 2.2. Accommodation facilities | 3.77 | 0.9895 |
| 2.3. Transportation facilities | 3.74 | 0.9816 |
| 2.4. Sport and recreation facilities | 2.81 | 0.7375 |
| 2.5. Food and drink facilities | 3.71 | 0.9738 |
| 2.6. Essential services | 3.42 | 0.8976 |
| **3. Enabling environment and authorities** | **3.26** | **0.8556** |
| 3.1. Public–private partnerships | 2.03 | 0.5328 |
| 3.2. Safety and security | 3.81 | 1 |
| 3.3. Government spending and efforts | 2.90 | 0.7612 |
| 3.4. Local leadership and political stability | 3.14 | 0.8241 |
| 3.5. Red tape limitation | 3.16 | 0.8294 |
| 3.6. Macro–economic environment | 2.64 | 0.6929 |

Source: Van der Schyff (2021) [32].

The index value was calculated by dividing each dimension and determinant with the largest average weighted value of 3.81 (safety and security) to produce an index value. For the dimensions and determinants to be on the same scale and to simplify interpretation, it is necessary to convert the average values to an index value. The higher the index value to one, the more important the determinant and dimension are to lead to tourism destination competitiveness. The index value calculation was required as the determinants and dimensions were weighted differently and should be on the same scale to ensure accurate analysis. The index value of each dimension and determinant was developed by use of the importance weights through the following formula:

Index value = determinant or group value/3.81 (largest weight value).

**Phase 6**: Pilot study: The pilot study was used to determine the validity and reliability of the measurement instrument (Hashim, Mukhtar & Safie, 2019) [38]. The testing of the measurement instrument was performed by using purposive sampling of 400 respondents in selected municipality districts in Sedibeng (Sample 1) consisting of three local municipalities and Fezile Dabi (Sample 2) consisting of four local municipalities. Before administering the questionnaire to samples 1 and 2, the purpose and objective of the research were clarified to the respondents and ensured confidentiality and anonymity and obtained consent from the respondents. Out of the 400 respondents, 197 were received from Sedibeng and 188 from Fezile Dabi district. As mentioned by Noar (2003) [58] and McGartland Rubio, Berg-Weger and Tebb, (2001) [59], certain researchers see a sample of 500 as desirable for performance exploratory or confirmatory analysis whereas other researchers regard 300 as sufficient and 150 as the minimum for the process of scale development. Table 5 provides the sample size and response rate.

**Table 5.** Sample size and response rate.

| Item | Sedibeng DM | Fezile Dabi DM | Total |
|---|---|---|---|
| Questionnaires distributed | 200 | 200 | 400 |
| Questionnaires returned | 197 | 188 | 385 |
| Unusable questionnaires | 19 | 28 | 47 |
| Useable questionnaires | 160 | 160 | 320 |
| Response rate | 98.5% | 94% | 96.25% |
| Percentage useable | 80% | 80% | 80% |

Source: Van der Schyff (2021) [32].

The primary purpose of this study was to develop an instrument that measures regional tourism destination competitiveness. The approach followed in this study was to use the Sedibeng district municipality's tourism industry as the pilot study (sample 1) to purify and refine the instrument, and sample 2: Fezile Dabi District Municipality, was used to replicate and test the results. The instrument administered consisted of 16 items that consisted of Section A, Demographical part (age, gender, district municipality, town or area of tourism activity, the respondent's area in tourism) and

Section B: Dimension 1: Resources, consisting of four items,
Section C: Dimension 2: Infrastructure, consisting of six items,
Section D: Dimension 3: Enabling Environment and Authorities consisting of six items.

*4.1. Data Analysis and Results*

The purpose of using SmartPLS in addition to SPSS is due to the additional test that could be conducted to analyse validity and reliability. To assess the factor analysis, KMO (Kaiser–Meyer–Olkin) Measure of Sampling Adequacy and Bartlett's Test Sphericity was used as given in Table 6 used to examine the appropriateness of factor analysis.

**Table 6.** Barlett's Test of Sphericity and KMO.

| | Sample 1: Sedibeng District Municipality | | | Sample 2: Fezile Dabi District Municipality | | |
|---|---|---|---|---|---|---|
| | Resources | Infrastructure | Enabling Environment and Authorities | Resources | Infrastructure | Enabling Environment and Authorities |
| Kaiser–Meyer–Olkin Measure of Sampling Adequacy | 0.747 | 0.757 | 0.839 | 0.752 | 0.750 | 0.769 |
| Bartlett's Test of Sphericity Approx. Chi–Square | 191.315 | 264.607 | 301.961 | 124.748 | 209.062 | 268.693 |
| Df | 10 | 21 | 21 | 10 | 21 | 21 |
| Sig. | 0.000 | 0.000 | 0.000 | 0.000 | 0.000 | 0.000 |

Source: Van der Schyff (2021) [32].

In Table 6 for sample 1, the KMO of O.747 for resources, a KMO of 0.757 for infrastructure and a KMO of 0.839 for enabling environment and authorities was obtained. In Table 6 for sample 2, a KMO of 0.752 for Resources, a KMO of 0.750 for infrastructure, and a KMO of 0.769 for Enabling Environment as indicated by Hair et al. (2010) [60] indicating that factor analysis would be appropriate.

Exploratory Factor Analysis for sample 1 and sample 2 performed to look at Bartlett's Test and Measurement of Sampling Adequacy. It is clear from Snedecor and Cochran (1989) [61] that non-normality would be shown if the sample originated from a non-normal distribution. As seen in Table 6, the test confirmed that factor analysis would indeed be applicable as the significance was below 0.05. Figures 1 and 2 provide the scree plots for the Sedibeng and Fezile Dabi district municipalities.

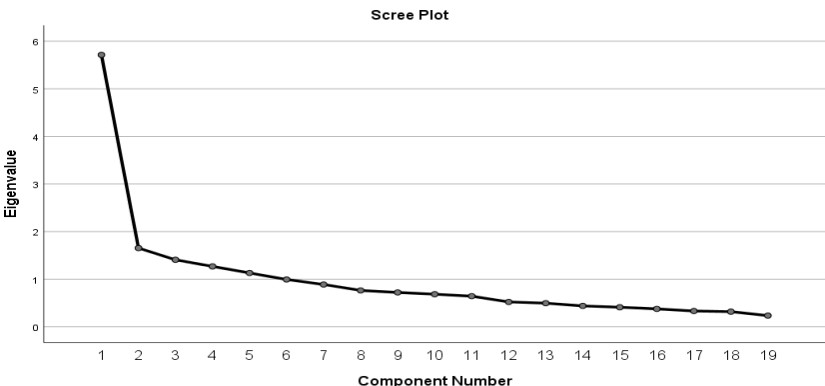

**Figure 1.** Scree plot for the Sedibeng district municipality. Source: Van der Schyff (2021) [32].

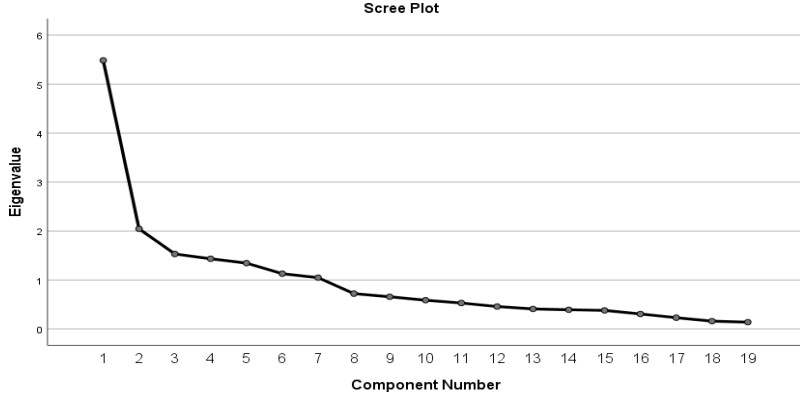

**Figure 2.** Scree plot for the Fezile Dabi district municipality. Source: Van der Schyff (2021) [32].

The scree plots also showed that three factors could be extracted for sample 1 and sample 2 Figure 1 (Sedibeng district municipality) and Figure 2 (Fezile Dabi district municipality). Table 7 gives the exploratory factor analysis for samples 1 and 2.

**Table 7.** Results of exploratory factor analysis (EFA) on the 16-items for the three dimensions.

| Item | Sample 1: Sedibeng District | | | | Sample 2: Fezile Dabi District | | | |
|---|---|---|---|---|---|---|---|---|
| | Factor Loading | Eigen Value | % Variance Explained | Cronbach Alpha | Factor Loading | Eigen Value | % Variance Explained | Cronbach Alpha |
| **Resources** | | 2.564 | 51.283 | 0.760 | | 2.287 | 45.748 | 0.694 |
| R1 | 0.765 | | | | 0.755 | | | |
| R2 | 0.754 | | | | 0.749 | | | |
| R3 | 0.719 | | | | 0.568 | | | |
| R4 | 0.622 | | | | 0.734 | | | |
| **Infrastructure** | | 3.027 | 43.249 | 0.778 | | 2.765 | 39.503 | 0.743 |
| I1 | 0.671 | | | | 0.616 | | | |
| I2 | 0.663 | | | | 0.662 | | | |
| I3 | 0.732 | | | | 0.551 | | | |
| I4 | 0.719 | | | | 0.697 | | | |
| I5 | 0.596 | | | | 0.632 | | | |
| I6 | 0.608 | | | | 0.663 | | | |
| **Enabling environment and authorities** | | 3.322 | 47.458 | 0.814 | | 2.928 | 41.823 | 0.763 |
| EA1 | 0.717 | | | | 0.594 | | | |
| EA2 | 0.600 | | | | 0.571 | | | |
| EA3 | 0.675 | | | | 0.600 | | | |
| EA4 | 0.678 | | | | 0.714 | | | |
| EA5 | 0.789 | | | | 0.661 | | | |
| EA6 | 0.655 | | | | 0.774 | | | |

Source: Van der Schyff (2021) [32].

Confirmatory Factor Analysis (CFA) was performed to assess validity using SmartPLS. Before SEM (structural equation modelling) can be performed the number of factors, and the item loadings onto the factor, need to be known. Structural validity of the scale was established. PLS–SEM was selected for the main repetition of the confirmatory analysis mainly because it fits non–normally scattered data (Henseler, Dijkstra, Sarstedt, Ringle, Diamantopoulos, Straub, Ketchen, Hair, Hult & Calantone, 2014) [62]. Therefore, CFA was completed as a second-factor analysis to enhance the assurance of a new instrument to measure tourism destination competitiveness from the viewpoint of respondents within the Sedibeng district municipality (sample 1) and the Fezile Dabi district municipality (sample 2).

To identify the dimensions/factors of samples 1 and 2, an EFA was performed to reduce the data and to refine the instrument and evaluate the discriminant validity of the dimensions/factors identified (Farrell, 2010) [63]. A simple principal component analysis was performed on 16 items for sample 1 and sample 2. To identify the dimensions or factors extracted, the eigenvalues, the percentage of variance explained, and individual factor loadings were deliberated. The results showed that three dimensions or factors were extracted with eigenvalues larger than one.

Table 7 identifies resources as a dimension as it has an eigenvalue above one. Therefore, the first component for Sedibeng district municipality explains 51.283 percent and for Fezile Dabi district municipality a 45.748 percent of the total variance, which is accepted in practice. It is clear from Table 7 that items did not cross-load and the factor loadings $\geq 0.4$ were considered significant. As portrayed, the factor loadings stretched from 0.569 to 0.789,

meaning that all items were useful measures of their factors. The Cronbach's Alphas that exceed 0.70 indicate that all the factors were internally consistent and well defined by their items (DeVellis, 2003) [46].

For the second dimension infrastructure for Sedibeng District municipality, 43.249 percent of the total variance is accounted for component 2 extracted and explained, but for Fezile Dabi district municipality total variance of 39.503. For the third dimension, a total variance of 47.458 percent can be explained for the Sedibeng district municipality and 41.823 per cent for the Fezile Dabi district municipality. The reliability statistics for the dimension resources are given in Table 7.

As seen in Table 7, the results of the Cronbach Alpha are for resources, infrastructure and enabling environment and authorities above 0.70 for samples 1 and 2, except for the Cronbach Alpha for Resources for sample 2 was below 0.70. However, as Nunnally and Bernstein (1994) [64] mentioned, it is acceptable. Therefore, the findings reported in Table 7 confirm the discriminant validity and the reliability of the 16 items used to measure the three dimensions for both sample 1 and sample 2. Both the discriminant validity and reliability demonstrate construct validity.

### 4.2. Assess Validity Using Confirmatory Factor Analysis (CFA)

Before SEM (structural equation modelling) can be performed, the number of factors, and the item loadings onto the factor needs to be known, and therefore, an EFA was completed before a CFA was performed. A PLS–SEM CFA was performed utilising SmartPLS software. Structural validity of the scale was established. PLS–SEM was selected for the primary iteration of the confirmatory analysis mainly because it is appropriate for non-normally distributed data (Henseler, Dijkstra, Sarstedt, Ringle, Diamantopoulos, Straub, Ketchen, Hair, Hult & Calantone, 2014) [62]. Therefore, CFA was performed for a second factor analysis to enhance the confidence of the new instrument to measure tourism destination competitiveness from the perspective of respondents within the Sedibeng district municipality (sample 1) and the Fezile Dabi district municipality (sample 2). Figures 3 and 4 indicate the results of the PLS–SEM confirmatory analysis.

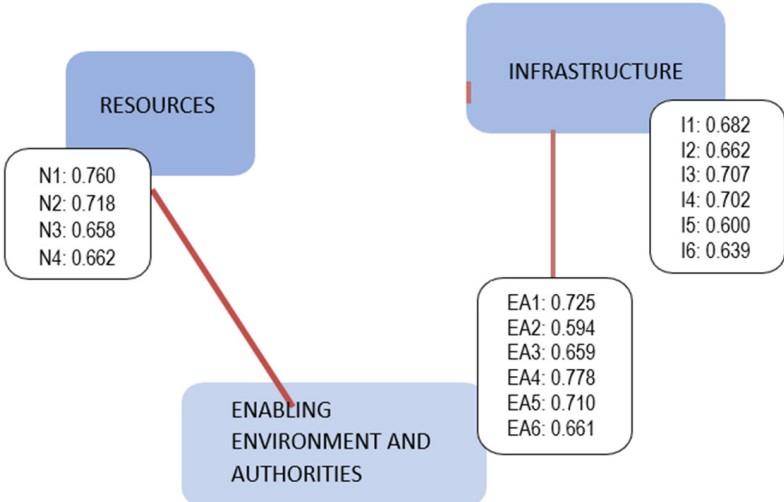

**Figure 3.** PLS–SEM confirmatory factor analysis for Sedibeng district municipality, with SmartPLS Source: Van der Schyff (2021) [32].

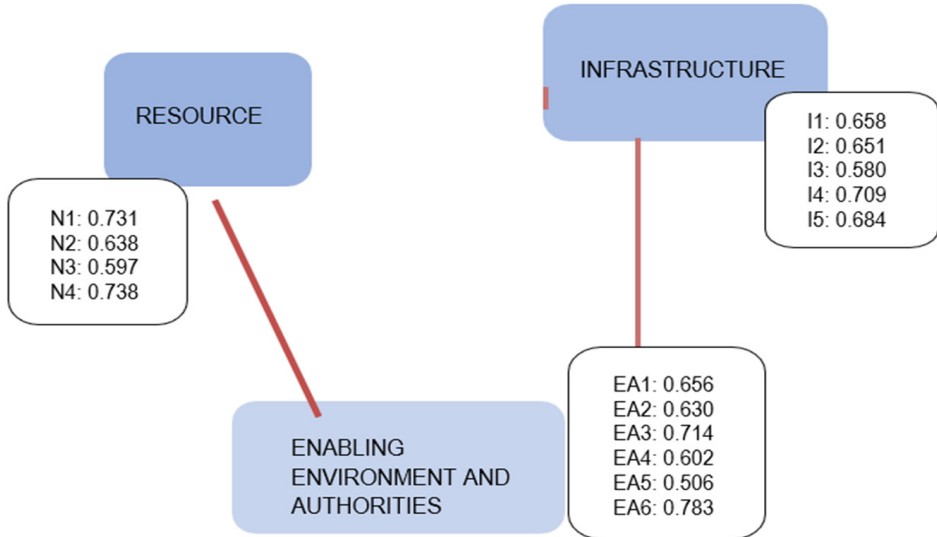

**Figure 4.** PLS–SEM confirmatory factor analysis for Fezile Dabi district, with SmartPLS. Source: Van der Schyff (2021) [32].

SmartPLS was used as the PLS-SEM extracted model is seen as a more true one according to the findings of Afthanorhan (2013) [65] who directed a cooperative CFA analysis using both SmartPLS and AMOS software and concluded that PLS-SEM path modelling using SmartPLS is appropriate to be utilised on the confirmatory factor analysis which is more reliable and valid and that is why PLS-SEM is used in this study as given in Table 8.

**Table 8.** PLS reliability and validity.

| Factor/Item | Sample 1: Sedibeng District Municipality | | | | Sample 2: Fezile Dabi District Municipality | | | |
|---|---|---|---|---|---|---|---|---|
| | Cronbach Alpha | CR | AVE | Rho_A | Cronbach Alpha | CR | AVE | Rho_A |
| Resources | 0.813 | 0.838 | 0.509 | 0.771 | 0.700 | 0.807 | 0.457 | 0.703 |
| Infrastructure | 0.778 | 0.840 | 0.430 | 0.780 | 0.743 | 0.818 | 0.393 | 0.752 |
| Enabling Environment & Authorities | 0.761 | 0.862 | 0.473 | 0.818 | 0.764 | 0.764 | 0.415 | 0.778 |

Source: Van der Schyff (2021) [32].

As seen in Table 8, the Cronbach Alpha results for sample 1 and sample 2 construct values are above 0.70, showing that the constructs are reliable. However, as Henseler, Ringle and Sinkovics (2009) [66] mentioned, Cronbach's Alpha can underestimate internal consistency reliability, which is why such a Composite Reliability (CR) can be more appropriate. As SmartPLS was used in the data analysis, composite reliability measure was checked to look at the internal consistency and as seen from the results above in Table 9 the values above are above 0.8 and 0.9 (Henseler et al., 2009) [66] and all values are considered as satisfactory. Only the value of enabling environment and authorities for a sample was just below 0.8, but was still satisfactory and as mentioned by Henseler et al. (2009) [66] only values under 0.6 show a lack of reliability.

**Table 9.** Discriminant validity.

| | Sample 1: Sedibeng District Municipality | | | Sample 2: Fezile Dabi District Municipality | | |
|---|---|---|---|---|---|---|
| | Resources | Infrastructure | Enabling Environment and Authorities | Resources | Infrastructure | Enabling Environment and Authorities |
| Resources | 0.676 | | | 0.713 | | |
| Infrastructure | 0.697 | 0.627 | | 0.641 | 0.656 | |
| Enabling environment and authorities | 0.658 | 0.719 | 0.644 | 0.564 | 0.650 | 0.688 |

Source: Van der Schyff (2021) [32].

To test for convergent validity, the AVE (Average Variance Extracted) value was used. AVE should be above 0.5 or more, and the CR 0.7 or more. CR (Composite Reliability) should be higher than the AVE (Götz, Liehr-Gobbers & Krafft 2010) [67]. However, as emphasised by Fornell and Larcker (1981) [68], even if AVE is less than 0.5, but composite reliability is higher than 0.6, the convergent validity of the construct is still adequate. As seen in Table 8, the obtained AVE values for infrastructure and authorities and enabling environment in sample 1 constructs were 0.430 and 0.473 and for sample 2 resources (0.457), infrastructure (0.393) and enabling environment and authority (0.415) respectively. When taken together with the values of CR, they were higher than 0.6 for each construct in samples 1 and 2, and it can be stated that convergent validity was established.

Table 8 depicts that all CR values are above 0.7, indicating internal consistency. All AVE are not above 0.5, indicating lack of convergent reliability. Finally, the values Rho_A reliability coefficients are all above 0.7, complying with the suggestions of Henseler, Dijkstra, Sarstedt, Ringle, Diamantopoulos, Straub, Ketchen, Hair, Hult and Calantone, (2014) [62]. Table 9 provides the discriminant validity for the Sedibeng and Fezile Dabi district municipalities.

Discrimination validity was assessed for both samples by comparing the square root of each AVE in the diagonal with the correlation coefficients (off-diagonal) for each construct in the relevant columns and rows. Step 1 is to prove that indicators strongly load more on their corresponding construct than on the other constructs, and the second step is comparing AVE value to inter-construct correlations. These square roots of AVE need to be larger than the inter-construct correlation (Chin & Newsted, 1999) [69]. As depicted in Table 9, there is discriminant validity between the constructs and is supported by the measurement model.

## 5. Conclusions

The development of the measurement instrument was explained as five phases beginning with the selection of determinants in the study of Van der Schyff (2021) [32], pre-testing and calculation of the index value results in the final tourism destination measurement instrument which can be applied to any regional tourism destination, followed by a statistical analysis of the measurement instrument on SPSS 26. The results signify that the tourism destination competitiveness measurement instrument is validated and can be applied to a tourism destination on a regional level.

The study's main objective was to develop a scale and to test the scale on two samples. The three-dimension and 16-items scales that were extracted using EFA have been validated. Statistical analysis of the tourism destination competitiveness measurement instrument applied to the Sedibeng and Fezile Dabi district municipalities. The statistical analysis was performed for each dimension and determinant of the measurement instrument for tourism destination competitiveness. The validity was confirmed, and Cronbach's Alpha confirmed reliability of the measurement instrument to be used as a measurement of tourism destination competitiveness.

This study contributes to the field of knowledge by developing a measurement instrument of a tourism destination's competitiveness as a facilitator of economic growth and

development. The measurement instrument serves as an empirical tool for regional tourism destination competitiveness, which enables the comparison of one tourism destination's performance or development to another tourism destination's performance or development. A limitation of the study is that, due to the fact that the tourism sector is multifaceted, there may be other determinates that have an influence on tourism development not listed in the instrument.

For future studies, the measurement instrument can be applied to tourism destinations to identify and compare the level of competitiveness between destinations on an empirical scale. This can assist these tourist destinations in the various elements that need attention and that need to be removed through strategy developments specifically for these destinations. This tool will assist policymakers/government organisations and tourism related businesses by identifying the strengths on which a region can build on, the opportunities which should be indevoured in order to increase the inflow of visitors and in return and increase in economic growth and development.

**Author Contributions:** Conceptualization, T.R. and D.F.M.; methodology, T.R.; software, T.R.; formal analysis, T.R.; investigation, T.R.; data curation, T.R.; writing—original draft preparation, T.R.; writing—review and editing, T.R. and D.F.M.; supervision D.F.M.; project administration, D.F.M.; funding acquisition, D.F.M. All authors have read and agreed to the published version of the manuscript.

**Funding:** This research received no external funding.

**Institutional Review Board Statement:** Not applicable.

**Informed Consent Statement:** Informed consent was obtained from all subjects involved in the study.

**Data Availability Statement:** Not applicable.

**Conflicts of Interest:** The authors declare no conflict of interest.

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
