# Peer review of "The Development of a Regional Tourism Destination Competitiveness Measurement Instrument"

_tourismhosp, doi:10.3390/tourhosp4010001_

Round 1

Reviewer 1 Report

The methods are consistent and suitable to the research question. The statistical analysis is detailed. However, several limitations are found:

a) The language and structure of the article seem more organized for a thesis than a scientific article. The article is very extensive, and a long part is dedicated to methodological explanations when it should focus on the contributions that this methodology can have for policy makers.

b) The whole structure should be simplified, e.g., the chapter “statistical analysis for instrument development”.

c) The literature review on the competitiveness of tourist destinations is very weak with outdated references.

d) The conclusions are poorly developed and the implications in terms of business, stakeholders and policy makers are weak.  

The main suggestion is to improve the paper in terms of scientific contribution and simplify all the structure.  The authors should state if other studies regarding this subject uses a different or similar methodology. Is it a new approach or is similar to the other studies in the field?  The findings are clear and objective, but can this methodology be applied to different destinations? What are the limitations of this study?

The paper needs substantial changes such as expanded review of literature and rewriting sections of the text, with substantial cuts in the justification of statistical analysis.

Author Response

COMMENTS OF REVIEWER ONE

ACTIONS TAKEN / RESPONSE TO REVIEWER(S)

The methods are consistent and suitable to the research question. The statistical analysis is detailed. However, several limitations are found:

Thank you, noted.

a) The language and structure of the article seem more organized for a thesis than a scientific article. The article is very extensive, and a long part is dedicated to methodological explanations when it should focus on the contributions that this methodology can have for policy makers

We followed the normal structure for an article.

The paper is only 9000 words, within the guidelines of the journal.

The methodology is an important part of this research paper as a new scale and measurement tools was development.

We added more regarding the contributions of the paper. This paper is ground breaking as this is a new regional tourism competitiveness instrument which could be used anywhere in the world.  

b) The whole structure should be simplified, e.g., the chapter “statistical analysis for instrument development”.

In Section 3.1.3 “statistical analysis for instrument development” is quite straight forward by introducing the test required to develop a tool or instrument that measured the development of a tourism region.

If the reviewer means that the rest of the sections should follow the same writing style (more simplistic), Section 4, was reduced as far possible to simplify reading.

c) The literature review on the competitiveness of tourist destinations is very weak with outdated references.

Section 2.3 was added to review the competitiveness of tourism destination with updated sources included.

d) The conclusions are poorly developed and the implications in terms of business, stakeholders and policy makers are weak.  

The following was added “This tool will assist policymakers/ government organisations and tourism related businesses by identifying the strengths on which a region can build on, the opportunities which should be indevoured in order to increase the inflow of visitors and in return and increase in economic growth and development.”

The main suggestion is to improve the paper in terms of scientific contribution and simplify all the structure. 

The authors should state if other studies regarding this subject uses a different or similar methodology. Is it a new approach or is similar to the other studies in the field? 

The findings are clear and objective, but can this methodology be applied to different destinations?

What are the limitations of this study?

The contribution of the study was given in Section 4 “This study contributes to the field of knowledge by developing a measurement instrument of a tourism destination's competitiveness as a facilitator of economic growth and development. The measurement instrument serves as an empirical tool for regional tourism destination competitiveness, which enables the comparison of one tourism destination's performance or development to another tourism destination's performance or development.”

The methodology used in the study is specific to the objectives of this study and included in Section 3.1.2. “This methodological approach is custom to the study’s objectives.” The following was added “This methodological approach is custom to the study’s objectives.”

Yes, the methodology can be applied to different destinations as stated in Section 2” When testing the measurement instrument by pilot studies in the regions, respondents were required to identify the level of importance of a tourism development determinant in a specific region”.  Section 4” For future studies, the measurement instrument can be applied to tourism desti-nations to identify and compare the level of competitiveness between destinations on an empirical scale.”

In addition Section 2.3 was added “The use of the tourism destination measurement instrument will be mainly as an indicator of which areas in a region can be used with opportunities of development, which are the strengths a region can build on, which are the weaknesses a region needs to minimize and any threats to development that should be anticipated in the future.”

The limitations of the study were added “A limitation of the study is, due to the fact that the tourism sector is multifaceted, there may be other determinates that have an influence on tourism development not listed in the instrument.” Future studies will test and confirm the factors as selected in different regions of the world.

The paper needs substantial changes such as expanded review of literature and rewriting sections of the text, with substantial cuts in the justification of statistical analysis.

1.        Section 2.3 was added for “the importance of tourism destination competitiveness”. Some sections were improved improved as requested. The statistical justification of test used in the analyses were provided In Section 3.1.3 eg “The factor analysis is a data reduction procedure. This test looks at the relationship between variables and identifies fewer variables than explaining these correlations or relationships. Factor analysis consists of two types, namely (i) principal component analysis and (ii) common factor analysis. The principal component analysis is the most frequently used test. Davies and Fearn (2004:1) [35] define PCA as a mathematical technique which modifies information “in a data set of samples”. This technique could be used for a few of many variables (Davies & Fearn, 2004:2) [35]. However, the more variables present, the more valuable the information”

“Factor analysis was used to determine the correlation between variables. Brandon (2011:48) [33] articulates that a correlation analysis is necessary to establish whether the variables are linked and in what way.”

“Cronbach Alpha is used to estimate or determine the internal consistency associated with the scores from a scale”

Reviewer 2 Report

The paper provides rather a general overview on various methods used in tourism development, however, the scientific justification of creating a measurement tool is not sound. Regarding the different factors contributing to tourism development, there is no literature review about the complexity of tourism and how different existing methods weighted the different factors. Literature review is rather on the methodology, detailing general concepts but it is not justified and explained how this method is able to reflect the regional level tourism competitiveness.

The conclusions are quite weak, the must be more based on the research results.

In addition to professional concerns, the paper needs some formatting development as well.

Overall, the paper has to be further developed, since based on the article, it is not proven that this instrument could be applied in any regions.

Author Response

COMMENTS OF REVIEWER TWO

ACTIONS TAKEN / RESPONSE TO REVIEWER(S)

The paper provides rather a general overview on various methods used in tourism development, however, the scientific justification of creating a measurement tool is not sound.

Regarding the different factors contributing to tourism development, there is no literature review about the complexity of tourism and how different existing methods weighted the different factors.

Literature review is rather on the methodology, detailing general concepts but it is not justified and explained how this method is able to reflect the regional level tourism competitiveness.

The tool was indicated to “This tool will assist policymakers/ government organisations and tourism related businesses by identifying the strengths on which a region can build on, the opportuni-ties which should be indevoured in order to increase the inflow of visitors and in return and increase in economic growth and development’ / “The development of an empirical measurement instrument is needed on a regional and local level, which will assist in comparing the region’s tourism development and competitiveness. The purpose of the research is therefore to develop an empirical measurement instrument that assists in the determination of a region’s tourism destination competitiveness.”

In section 2.3 the tourism-led growth hypothesis was indicated which provides the importance of tourism for the economy

/ 3. Section 2.3 was added for “the importance of tourism destination competitiveness”.

This study builds on the previous study van der Schyff (2021) where an extensive literature review was conducted on the various determinants of tourism destination competitiveness.

The review of methods used to develop scales (instruments) is highlighted to indicate the manner is which a instrument is development. In the results section the validity and reliability test indicate that the measurement instrument can be used to show the level of development of a tourism destination

The conclusions are quite weak, the must be more based on the research results.

Section 4 was updated accordingly

In addition to professional concerns, the paper needs some formatting development as well.

Noted and improved.

 Overall, the paper has to be further developed, since based on the article, it is not proven that this instrument could be applied in any regions.

The results in Section die validity and reliability test indicate that the measurement instrument can be applied to regions in order to analyse their level of competitiveness / development.

Reviewer 3 Report

The paper says that "limited evidence was found of regional tourism destination competitiveness 28 measurement tools." But is that really the case? What about these papers? 

Croes, Robertico. "Measuring and explaining competitiveness in the context of small island destinations." Journal of travel research 50.4 (2011): 431-442.

McLennan, Char-Lee J., et al. "Developing and testing a suite of institutional indices to underpin the measurement and management of tourism destination transformation." Tourism Analysis 18.2 (2013): 157-171.

Mendola, Daria, and Serena Volo. "Building composite indicators in tourism studies: Measurements and applications in tourism destination competitiveness." Tourism Management 59 (2017): 541-553.

Van der Schyff, Tanya, and Daniel F. Meyer. "The formulation of a regional tourism destination competitiveness measurement instrument." Journal of Public Administration 54.4-1 (2019): 873-887.

Zins, Andreas H. "Internal BenchmarkIng for regIonal tourIsm organIzatIons: a case example." Tourism Analysis 19.4 (2014): 413-424.

Krešić, Damir, and Darko Prebežac. "Index of destination attractiveness as a tool for destination attractiveness assessment." Tourism: An International Interdisciplinary Journal 59.4 (2011): 497-517. 

Dwyer, Larry, et al. "Attributes of destination competitiveness: A factor analysis." Tourism analysis 9.1-2 (2004): 91-101.

The literature review of this review paper seems woefully inadequate, and as such it is unclear what is new or why it should be considered significant. 

Author Response

COMMENTS OF REVIEWER THREE

ACTIONS TAKEN / RESPONSE TO REVIEWER(S)

The paper says that "limited evidence was found of regional tourism destination competitiveness 28 measurement tools." But is that really the case? What about these papers? Croes, Robertico. "Measuring and explaining competitiveness in the context of small island destinations." Journal of travel research 50.4 (2011): 431-442.

 McLennan, Char-Lee J., et al. "Developing and testing a suite of institutional indices to underpin the measurement and management of tourism destination transformation." Tourism Analysis 18.2 (2013): 157-171.

 Mendola, Daria, and Serena Volo. "Building composite indicators in tourism studies: Measurements and applications in tourism destination competitiveness." Tourism Management 59 (2017): 541-553.

 Van der Schyff, Tanya, and Daniel F. Meyer. "The formulation of a regional tourism destination competitiveness measurement instrument." Journal of Public Administration 54.4-1 (2019): 873-887.

 Zins, Andreas H. "Internal BenchmarkIng for regIonal tourIsm organIzatIons: a case example." Tourism Analysis 19.4 (2014): 413-424.

 Krešić, Damir, and Darko Prebežac. "Index of destination attractiveness as a tool for destination attractiveness assessment." Tourism: An International Interdisciplinary Journal 59.4 (2011): 497-517. 

 Dwyer, Larry, et al. "Attributes of destination competitiveness: A factor analysis." Tourism analysis 9.1-2 (2004): 91-101.

Majority of these studies focusses on a conceptual model or measurement by simply identifying the determinants that enables tourism competitiveness in a region.

Our study takes an empirical measurement approach.

The sentence was amended to state this “In reviewing the literature, is shown that there is still a need for research in regional tourism destination competitiveness as there is no extensive evidence was found of regional tourism destination competitiveness measurement tools in an empirical form.”

The literature review of this review paper seems woefully inadequate, and as such it is unclear what is new or why it should be considered significant. 

The literature of this paper focus on the development of the instrument. Therefore, most literature can be found in section 2.2. “Best practice principles in scale (measurement instrument) development”. Section 2.3 was added for “the importance of tourism destination competitiveness”. “Within the global economy, competitiveness is increasingly becoming a requirement to remain successful (Luˇstický & ˇStumpf, 2021:2) [55] (Shariffuddin Azinuddin, Hanafiah & Zain, 2021:1) [56]. This is not different for tourism destinations (regions). In order for a tourism destination to remain competitiveness is should (i) ultimately attract tourist and/or visitors (ii) grow with international globalisation (iii) provide a unique experience (Shariffuddin Azinuddin, Hanafiah & Zain, 2021:2) [56].

The tourism competitiveness of a region could lead to a range of benefits. Rodríguez, Florido and Jacob (2020:2) [54] agree that the tourism sector is an important contributor to economic growth and job creation. Infrastructure development and tax generation are amongst the economic benefits of an increase in tourism development (Cavalheiro, Joia & Cavalheiro, 2020:) [57].  In addition, economic development is known to be a benefit for tourism develpment (Rodríguez, Florido & Jacob, 2020:3) [54]. The increase in technology, skills development and higher human capital contribute to economic development in a region.

According to Madanaguli, Srivastava, Ferraris and Dhir (2022:447), [53] in recent years countries focused not only on the financial gains of economic activities but also focusing on the environmental aspect thereof and as such sustainable tourism development is a key objective of tourism destinations. They main negative consequence of extreme tourism develop is resource depletion (Rodríguez, Florido & Jacob, 2020:3) [54]. Some regions do not have the capacity to compensate for the increase rise in economic and social activities.

Social Corporate Responsibility comes into play as companies and regions takes into account how business practices, policies, growth and enhancements influence the region’s environment (Madanaguli, Srivastava, Ferraris & Dhir (2022:447) [53]. The two key components that are influences by tourism development is the environment and the community members. Therefore changes, enhancements and improvements of the region should take into account the influence it will have on the environment and community members.

The tourism-led growth hypothesis states the importance of tourism development in the growth of a region’s economy (Xia, Doğan, Shahzad, Adedoyin, Popool & Bashir, 2022:5) [58]. According to Pérez-Montiel, Asenjo and Erbina (2021:2) [59], the tourism-led growth hypothesis explaines that the increase in tourism development leads to an increase in economic growth. This theory has been proven by various studies (Balaguer & Cantavella-Jordá (2002) [60]; Pérez-Montiel, Asenjo & Erbina, 2021 [59]) which adds to its validity as a premise for this study. The use of the tourism destination measurement instrument will be mainly as an indicator of which areas in a region can be used with opportunities of development, which are the strengths a region can build on, which are the weaknesses a region needs to minimize and any threats to development that should be anticipated in the future. The end goal is to increase tourism development of a region keeping in mind the needs of the environment and community members.”

In Section 2 it is states that “the validation of a measurement instrument can be complicated.” And therefore, it is important to focus on the various methods of scale development.

In section 2.1 the following was added ”Due to the complexity of scale development, numerous methods or techniques can be utilised in developing a tourism destination measurement instrument (scale).”

In addition, we added a paragraph in Section 2.2 explaining why the four studies of Churchill, Hinkins, Rossiter and DeVellis and Worthington and Whittaker’s was used.

Round 2

Reviewer 1 Report

The authors should simplify, as stated earlier, the methodological analysis and particularly the justification of the statistical options and interpretations. As instance, the readers don´t need to know the explication of what is a factor analysis, they just need to know why do you choose this technique and why is more important then the others to measure competitiveness. The validity of the indicators to analyse the quality of the factor analysis is important to maintain in the paper.   

Author Response

Reviewer comments

Action/ response

Reviewer 1

The authors should simplify, as stated earlier, the methodological analysis and particularly the justification of the statistical options and interpretations.

As instance, the readers don´t need to know the explication of what is a factor analysis, they just need to know why do you choose this technique and why is more important then the others to measure competitiveness.

The validity of the indicators to analyse the quality of the factor analysis is important to maintain in the paper. 

Noted, Thank you.

Sections 3 and 4 reduced and minimalised.

The importance of the different types of analyses were highlighted.

Reviewer 2 Report

The authors revised the paper and in this form, it is suitable for publishing.

Author Response

Reviewer 2

The authors revised the paper and in this form, it is suitable for publishing.

Noted, Thank you.

Reviewer 3 Report

Thank you for an improved draft! Unfortunately, however, the paper's literature review remains lacking. The authors claim that "there is no extensive evidence was found of regional tourism destination competitiveness measurement tools in an empirical form.” That appears to be empirically false. See: 

https://scholar.google.com/scholar?hl=en&as_sdt=0%2C9&q=+regional+tourism+empirical+measurement&btnG=

Author Response

Reviewer 3

Thank you for an improved draft!

 Unfortunately, however, the paper's literature review remains lacking.

The authors claim that "there is no extensive evidence was found of regional tourism destination competitiveness measurement tools in an empirical form.”

That appears to be empirically false

https://scholar.google.com/scholar?hl=en&as

_sdt=0%2C9&q=+regional+tourism+empirical +measurement&btnG=

Noted, Thank you.

The statement made previously was changed to “In reviewing the literature, it is shown that there is still a need for research in regional tourism destination competitiveness as there is no extensive evidence was found of regional tourism destination competitiveness measurement tools in an empirical form”

The study (previous study) “The development and testing of a measurement instrument for regional tourism competitiveness facilitating economic development” found in the link as provided, is the basis on which this current study is done in terms of determinants identification (literature review). This study as listed in the link was done by us, Meyer and Rheeders. 

The other studies have a more narrow focus on concepts such as (i) “sustainability” which uses sustainability paradigms and (ii) social network analysis to measure institutional thickness in tourism destinations.

Some using other methods such as PROMETHEE approach. Our measuring instrument tries to include all components of regional tourism competitiveness.

As such the authors believe that the study still has merit and importance.

Round 3

Reviewer 3 Report

It seems that there may be an undervaluing of the literature review and an overstatement of the importance of this work. It is not that this work is either the first of its kind or it is meaningless. This work can still have "merit and importance" while acknowledging the previous similar work that has been done. Indeed, this work will have more importance if the manuscript can situate itself  better in the body of existing literature. So rather than scoff at the extensive existing body of similar studies, it would be infinitely better if this paper would engage with that extensive body, discuss in detail what can be learned from previous work and what sets this study as different. It is the literature review, if done well, that justifies the importance of this new work. But there seems to be great hesitancy to actually engage with much of the similar work that already has been done in this area. 

Author Response

Reviewer 3

Comments/ actions

It seems that there may be an undervaluing of the literature review and an overstatement of the importance of this work.

 It is not that this work is either the first of its kind or it is meaningless. This work can still have "merit and importance" while acknowledging the previous similar work that has been done.

Indeed, this work will have more importance if the manuscript can situate itself better in the body of existing literature.

This research project consists of three papers, with paper 2 the current one under discussion:

·       Paper 1 (Literature and empirical

·       review of the determinants of tourism destination competitiveness), published 

·       Paper 2 (The development of a regional tourism destination competitiveness measurement instrument) current manuscript) current paper

·       Paper 3 (The application of the regional tourism destination competitiveness measurement instrument to the Sedibeng ad Fezile Dabi district municipality), in the process of being formulated

The authors believe that paper 2 has an adequate review of literature as it builds on the previous paper (paper 1) i.t.o a literature review with regards to the determinants/factors/elements that contribute to tourism destination competitiveness. In order to not duplicate work, the decision was made to refer to the previous study.

Therefore, the current paper (paper 2) focus primarily on the development process of a measurement instrument and presents the scale development process in Section 2.1. and the Best practices in scale development in Section 2.2. to provide an understanding of instrument development and previous methods used.

The authors did add Section 2.3. for the importance of tourism destination competitiveness to increase the strength of the literature review.

So rather than scoff at the extensive existing body of similar studies, it would be infinitely better if this paper would engage with that extensive body, discuss in detail what can be learned from previous work an what sets this study as different.

It is the literature review, if done well, that justifies the importance of this new work. But there seems to be great hesitancy to actually engage with much of the similar work that already has been done in this area

Majority of work done in this area is explained in manscript 1, however a new section was added (Section 2.3) to provide an overview of previous studies that aimed to develop a scale or measurement instrument of tourism destination competitiveness.

Study

Similar to current study

Different to current study

Ritchie and Crouch (2010)

Goal:

Measuring TDC on a model

Takes into account the global (macro) environment and competitive (micro) environment

Method:

Qualitative interviews

Hanafiah, Hemdi and Ahmad (2016)

Goal:

Develop a model for TDC

Goal:

the goal was to develop a performance-based conceptual framework/ model of TDC based on competitiveness theory.

Method:

Conceptual study – literature review

Selim, Abdel-Fattah and Hegazi, (2021)

Method:

Survey approach (Pilot study)

Purposive sampling

Exploratory Factor Analysis

Confirmatory Factor Analysis

Method:

Mix method study

Goal:

Investigating the attractiveness of heritage destinations through a composite index

Sul, Chi and Han (2022)

Method:

Empirical method

Survey approach

Convenience sampling for experts (tourism-related managers

Exploratory Factor Analysis

Confirmatory Factor Analysis

Goal:

Sole focus on the business environment by aiming toward competitive advantage
